# Prolyl Isomerase, Pin1, Controls Meiotic Progression in Mouse Oocytes

**DOI:** 10.3390/cells11233772

**Published:** 2022-11-25

**Authors:** Yumi Hoshino, Takafumi Uchida

**Affiliations:** 1Laboratory of Animal Reproduction, Graduate School of Integrated Science for Life, Hiroshima University, Hiroshima 739-8528, Japan; 2Laboratory of Reproductive Biology, Faculty of Science, Japan Women’s University, Tokyo 112-8681, Japan; 3Laboratory of Molecular Enzymology, Department of Molecular Cell Science, Graduate School of Agricultural Science, Tohoku University, Miyagi 981-8555, Japan

**Keywords:** oocyte, meiotic maturation, Pin1, phosphorylation, mouse

## Abstract

During meiotic maturation, accurate progression of meiosis is ensured by multiple protein kinases and by signal transduction pathways they are involved in. However, the mechanisms regulating the functions of phosphorylated proteins are unclear. Herein, we investigated the role of Pin1, a peptidyl-prolyl cis-trans isomerase family member that regulates protein functions by altering the structure of the peptide bond of proline in phosphorylated proteins in meiosis. First, we analyzed changes in the expression of Pin1 during meiotic maturation and found that although its levels were constant, its localization was dynamic in different stages of meiosis. Furthermore, we confirmed that the spindle rotates near the cortex when Pin1 is inhibited by juglone during meiotic maturation, resulting in an error in the extrusion of the first polar body. In Pin1^−/−^ mice, frequent polar body extrusion errors were observed in ovulation, providing insights into the mechanism underlying the errors in the extrusion of the polar body. Although multiple factors and mechanisms might be involved, Pin1 functions in meiosis progression via actin- and microtubule-associated phosphorylated protein targets. Our results show that functional regulation of Pin1 is indispensable in oocyte production and should be considered while developing oocyte culture technologies for reproductive medicine and animal breeding.

## 1. Introduction

The resumption of meiosis after diplotene arrest, progression through metaphase I (MI), and re-arrest at metaphase II (MII) are essential for meiotic maturation in mammalian oocytes. These processes are controlled by cyclic changes in the activity of cyclin-dependent kinase 1 as well as by cyclin B1 degradation, which lead to stabilization/destabilization of the maturation-promoting factor [1,2]. Oocytes acquire fertilization and developmental competence by synchronizing nuclear and cytoplasmic maturation during meiosis. Nuclear maturation involves germinal vesicle (GV) breakdown (GVBD) and assembly of the meiosis I spindle. Eccentric positioning of the spindle leads to asymmetric cell division, which coincides with a reduction in the number of chromosomes through the extrusion of the first polar body and maximal retention of the cytoplasm of oocytes [3]. These events occurring during oocyte maturation are regulated by multiple kinases [4,5,6], particularly serine or threonine (Ser/Thr) protein kinases [7]. Moreover, when oocytes are exposed to intracellular signals after stimulation with endogenous luteinizing hormone, meiosis resumes selectively and stops again at MII to maintain the fertilization ability.

In a previous study, we reported the role of Akt (protein kinase B), a Ser/Thr protein kinase, in the formation of the meiotic spindle and the extrusion of the second polar body in mouse oocytes [6]. Additionally, another Ser/Thr kinase, the mammalian target of rapamycin (mTOR), forms a complex with raptor or rictor—the mTOR/raptor complex controls spindle function and the mTOR/rictor complex contributes to actin-dependent asymmetric division during meiotic maturation in mice [8]. Many proteins are expressed and phosphorylated constantly during oocyte maturation [9,10]. As a result, a single protein can simultaneously perform different and pleiotropic roles, as observed for Akt and mTOR. However, the mechanisms regulating the functions of phosphorylated proteins remain unclear.

In somatic cells, protein function is regulated by phosphorylation as well as by structural changes in proteins occurring after phosphorylation. The proline isomerase (enzyme peptidyl-prolyl cis-trans isomerase) family promotes the proper folding of proteins and regulates their functions by altering the structure of the peptide bond formed by proline [11]. Pin1, a proline isomerase family member, is a unique enzyme that specifically isomerizes phospho-Ser-Pro or phospho-Thr-Pro (pSer/pThr-Pro) peptide bonds [12], thereby, regulating the functions, stability, and localization of phosphorylated proteins carrying the pSer/pThr-Pro motifs. Moreover, Pin1 is phosphorylated at multiple pSer/pThr-Pro motifs and induces mitotic progression in eukaryotic cells [13]. Pin1, in turn, also acts on several mitosis-specific phosphoproteins [14]. Although reversible phosphorylation of proteins in the context of pSer/pThr-Pro is critical in the regulation of numerous cellular events [15], the role of Pin1 in meiotic maturation is unclear. 

In this study, to clarify the role of Pin1 in meiotic maturation, we analyzed the changes in the intracellular localization of this protein. We also analyzed the localization dynamics of phosphorylation targets of Pin1 using mitotic protein monoclonal 2 antibody (MPM2), which can detect most of the mitotic proteins phosphorylated at the Ser/Thr-Pro residues [14], and the effect of Pin1 inhibition on oocyte meiotic maturation. 

## 2. Materials and Methods

### 2.1. Animals

C57BL/6j mice were purchased from Charles River Laboratories, Inc. (Kanagawa, Japan). Pin1^−/−^ mice were generated and bred according to the methods described previously [16]. The mice were bred by mating Pin1^−/−^ with wild-type or Pin1^+/−^ mice, because Pin1^−/−^ mice had a profound fertility defect. The mice bred by mating Pin1^−/−^ or Pin1^+/−^ mice were genotyped using polymerase chain reaction. The primers used were as follows: forward, 1.2L (5′-GCACCCGATCCTGTTCTGGAAACTCAG-3′); reverse, Wild1.2A (5′-CATGAGAAGGGATTAGAAGCAAGATTCGACTGG-3′); reverse, Start2A (5′-GCCAGAGGCCACTTGTGTAGCGC-3′).

### 2.2. Oocyte Collection and In Vitro Maturation

Collection and in vitro maturation (IVM) of oocytes were performed as described previously [6,17]. Oocytes collected from the ovaries were cultured after the removal of cumulus cells in Waymouth’s medium (cat. No. 11220035; Thermo Fisher Scientific, Waltham, MA, USA) containing 4 mM hypoxanthine (cat. No. H9377; Sigma-Aldrich, St. Louis, MO, USA), to suppress spontaneous maturation. To obtain oocytes from all the maturation stages, they were collected at 0, 4, 6, 8, 10, 12, and 18 h after the start of culture. In vivo-matured oocytes were collected from the oviduct at 15 and 24 h after injection of human chorionic gonadotropin (hCG; ASKA Animal Health, Tokyo, Japan) as “fresh” and “aged” oocytes, respectively [18]. The collected oocytes were subjected to immunofluorescence staining or immunoblotting analysis. In all experiments, 8 to 10 mice were used and oocytes were randomly divided for each condition.

### 2.3. Pin1 Inhibition

For inhibition of Pin1 during meiotic maturation, juglone (cat. No. H47003; Sigma-Aldrich, St. Louis, MO, USA) was added to the culture medium. A 100 mM stock solution of juglone was prepared in dimethyl sulfoxide and diluted in culture medium to final concentrations of 5 and 10 µM.

### 2.4. Immunoblotting Analysis

Oocytes were collected and placed in sample buffer containing 2-mercaptoethanol for sodium dodecyl sulfate-polyacrylamide gel electrophoresis (cat. No. 30566-22; Nacalai Tesque, Kyoto, Japan). Lysates were separated by electrophoresis (cat. No. 4561035; Bio-Rad, Hercules, CA, USA) and the resolved proteins were transferred onto polyvinylidene difluoride membranes (cat. No. 1704156; Bio-Rad). The membranes were incubated with the following antibodies: anti-Pin1 (cat. No. 3722; 1:1000; Cell Signaling Technology, Danvers, MA, USA), anti-MPM2 (cat. No. 05-368; 1:500; Millipore, Billerica, MA, USA), anti-phospho-Akt (Ser473) (cat. No. 9271; 1:1000; Cell Signaling Technology), anti-phospho-mTOR (Ser2448) (cat. No. 5536; 1:1000; Cell Signaling Technology), and anti-β-actin (cat. No. ab8227; 1:1000; Abcam, Cambridge, UK) overnight at 4 °C. Anti-MPM2 is a monoclonal antibody for detection of phospho-Ser/Thr-Pro. Antigens were detected using the ECL Prime Western blotting detection reagent (cat. No. RPN2232; GE Healthcare, Little Chalfont, UK) and the Luminescent Image Analyzer, LAS-3000 (Fujifilm Life Science, Tokyo, Japan). Densitometric analysis of the blots was performed by measuring the intensity of each band using the ImageJ software (NIH, Bethesda, MD, USA), and the data were normalized against the relevant β-actin loading controls.

### 2.5. Immunofluorescence Staining

Oocyte fixation and immunofluorescence staining were performed as previously described [6,17]. Pin1 and MPM2 were detected using anti-Pin1 (cat. No. sc-15340; 1:100; Santa Cruz Biotechnology, Dallas, TX, USA) and anti-pSer/pThr-Pro MPM2 antibodies (cat. No. 05-368; 1:100; Millipore). The secondary antibodies used were Alexa Fluor 488-conjugated anti-rabbit IgG (cat. No. A11034; 1:200; Invitrogen, Carlsbad, CA, USA) to detect Pin1 and Alexa Fluor 568-conjugated anti-mouse IgG (cat. No. A11004; 1:200; Invitrogen) to detect MPM2. Microtubules were detected using anti-α-tubulin antibodies (cat. No. T9026; 1:500; Sigma-Aldrich) and Alexa Fluor 488-conjugated anti-mouse IgG (cat. No. A11001; 1:200; Invitrogen) or Cy5-labeled anti-mouse IgG (cat No. 115-175-147; 1:200; Jackson ImmunoResearch, West Grove, PA, USA). F-actin was detected using Acti-stain 488 Fluorescent Phalloidin (cat no. PHDG1; 1:100; Cytoskeleton, Denver, CO, USA). To detect Akt, phospho-Akt, mTOR, and phospho-mTOR, we used anti-Akt, phospho-Akt (Ser473), mTOR, and phospho-mTOR (Ser2448) antibodies (Cell Signaling Technology; cat. No. 9272, 9271, 2983, 5536, respectively, 1:200). Chromosomes were labeled with 4′,6-diamidino-2-phenylindole (cat. No. D1306; 1:1000; Invitrogen) or propidium iodide (cat. No. 341-0788; 10 µg/mL; Wako, Osaka, Japan). Fluorescence intensity was quantitated using the ImageJ software. Each experiment was performed five times, and more than 100 oocytes were used for each condition. The oocytes were imaged using a Zeiss LSM700 confocal scanning laser microscope and the images were analyzed using the ZEN software.

### 2.6. Statistical Analysis

Experiments were performed in three or more independent biological replicates. The data are presented as the percentage or mean ± standard error of the mean (SEM). Statistical significance was determined by using the one-way analysis of variance (ANOVA), followed by Fisher’s protected least significant difference test, with *p* < 0.05 considered statistically significant.

## 3. Results

### 3.1. Expression of Pin1 Remains Constant during Meiotic Maturation, but That of MPM2 Changes with Meiotic Progression

To investigate the changes in the expression of Pin1 and phospho-mitotic proteins detected using anti-MPM2 during meiotic maturation, proteins were extracted from oocytes cultured for 0, 4, 6, 8, 10, 12, or 18 h and from oocytes collected from the oviducts at 15 and 24 h after hCG injection. The oocyte stages used for immunoblotting analysis are shown in Figure 1A. The incidence of GVBD was observed in 5.7%, 39.3%, 73.3%, and 100% oocytes collected at 4, 6, 8, and 10 h, respectively. The percentage at 10 h also included oocytes reaching MI. The percentage of oocytes confirmed to be in MII, by measuring polar body extrusion, was 17.0% at 12 h, and polar body extrusion was observed in all oocytes at 18 h. 

The time course of expression of Pin1 and proteins detected using anti-MPM2 during meiotic maturation is shown in Figure 1B. The expression of Pin1 was constant throughout maturation. In contrast, the expression of phosphorylated mitotic proteins varied greatly during maturation. At 8 h, some oocytes may have reached MI. Several bands were detected for MPM2 after 8 h, suggesting that there were multiple Pin1 targets during maturation.

### 3.2. Localization of Pin1 Is Altered during Maturation of Mouse Oocytes

First, to elucidate the involvement of Pin1 in oocyte maturation, its localization across various stages of maturation was determined (Figure 2A–D). To detect the localization of phosphorylation targets of Pin1, changes in the MPM2 antibody staining, as a marker of phosphorylated Ser/Thr-Pro, were examined. Fully grown GV-stage oocytes were collected from the ovarian follicles of 3-week-old wild-type (WT) mice and cultured for 18 h in Waymouth’s medium containing 4 mM hypoxanthine. A representative localization pattern based on immunostaining of more than 100 oocytes, fixed over time, showed the characteristic behavior of Pin1 and phosphorylated mitotic proteins in GV, GVBD (also called pro-metaphase I [PMI]), MI, anaphase/telophase I (A/TI), and MII (Figure 2A). In GV-stage oocytes, Pin1 was detected throughout the cytoplasm but showed particularly strong nuclear localization. The expression of proteins detected using the anti-MPM2 antibody was observed in the nucleus as spots (0 h GV) and was localized in the vicinity of the nuclear membrane, 4 h after the start of culture (4 h GV). In GV-stage oocytes, observed at 6 h (6 h GV) after IVM, Pin1 was localized around the nuclear membrane, and strong MPM2 staining was observed at the nuclear membrane. These results suggest that Pin1 or its related factors are involved in the collapse of the nuclear envelope. As shown in Figure 2A (for 6 h PMI) and Figure 2B, after GVBD, the expression of MPM2 antigens increased dramatically, and they became strongly localized in the cytoplasm (6 h PMI). The changes in the expression of proteins detected using the anti-MPM2 antibody at GV and after GVBD are shown in Figure 2B. The fluorescence signal for MPM2 was stronger in oocytes that had undergone GVBD than in the GV-stage oocytes. In contrast, Pin1 was localized throughout the cytoplasm and was strongly expressed around the chromosomes (8 h PMI). During MI, A/TI, and MII, Pin1 and MPM2 showed similar localization patterns (10 h MI, 12 h A/TI, 18 h MII). Next, we examined the MPM2 localization in metaphase oocytes, using confocal Z-stacks. Based on an optical section showing the spindle position, we found that anti-MPM2-reactive phospho-mitotic proteins were strongly localized in the MI oocyte cortex (Figure 2C). Additionally, Pin1 was expressed throughout the cytoplasm and was enriched at the spindle. Interestingly, Pin1 was localized in the midbody, forming a contractile ring during the later stages of division (Figure 2A, 12 h A/TI). During oocyte maturation in vertebrates, cytokinesis is initiated after one pole of the bipolar MI spindle attaches to the oocyte cortex, resulting in the formation of a polar body and a mature oocyte [7,19,20,21]. The essential role of a properly formed direct contractile ring in separating the cytoplasm and polar body during polar body extrusion and the observed Pin1 localization suggest that Pin1 is likely involved in division.

To observe changes in the localization of Pin1 and anti-MPM2-reactive phospho-mitotic proteins due to the aging of oocytes, 3-week-old WT mice were administered hCG, 46 h after injecting pregnant mare’s serum gonadotropin, and oocytes were collected from the oviduct at 15 or 24 h. The oocytes were then immunostained with anti-Pin1 and anti-MPM2 antibodies. Pin1 was localized at the spindle, 15 h after hCG injection, although after 24 h, it was re-localized to the chromosomes (Figure 2D). Oocyte arrest was observed in the MII stage until fertilization, after extrusion of the first polar body; however, oocytes undergo aging over time. Typical changes in aged oocytes included elongation of the meiotic spindle and spreading of the spindle poles (hCG 24 h type I). Chromosome separation or misalignment was also observed (hCG 24 h type II) [18,22]. These results suggest that Pin1 is expressed on the meiotic spindle only in the younger oocytes with developmental competence and the localization changes with aging.

### 3.3. Juglone Inhibits the Extrusion of the First Polar Body in Mouse Oocytes

The effects of inhibition of Pin1 activity were assessed using the Pin1-specific inhibitor, juglone (5-hydroxy-1,4-naphthoquinone). The level of Pin1 decreased with the administration of juglone in a concentration-dependent manner; at 10 µM, the expression was approximately one-fourth of that in the control (Figure 3A; 51% and 25% expression at 5 and 10 µM, respectively, compared with that in control). Although higher concentrations of juglone may block Pin1 in oocytes, in previous studies, we observed that treatment of oocytes with 25 µM or higher concentrations of juglone for 18 h resulted in 0% progression to MII (unpublished data). It has been confirmed that the frequency of nuclear condensation also increases. In this study, the effects of 5 and 10 µM juglone were compared, as no cell death occurred at these concentrations. The maturation rates (at the MII stage) after 18 h were 87.2%, 71.9%, and 34.7% in oocytes treated with vehicle control, 5 µM juglone, and 10 µM juglone, respectively, indicating that juglone inhibits the extrusion of the first polar body (Figure 3B). Oocytes in MI showed abnormalities in spindle morphology or chromosome localization. However, chromosome condensation and cell death were not observed (Figure 3C). After 18 h of culture, the oocytes were mature and exhibited an intact spindle and the first polar body, as observed in the control. However, in the treated group, 28.1% and 65.3% of oocytes were arrested at MI, following treatment with 5 and 10 µM juglone, respectively, and showed disrupted spindle fibers, and undefined spindle poles (Figure 3Ca–c). The MI spindles were rotated and positioned parallel to the oocyte cortex (Figure 3Ca). Additionally, the spindle was small, chromosomes were scattered away from the metaphase plate, the polar body was not extruded (Figure 3Cb), and chromosome separation was evident in the cytoplasm (Figure 3Cc). These results suggest that Pin1 is involved in the formation and maintenance of the spindle and extrusion of the first polar body in oocytes. The oocytes were classified into three types, namely, A, B, and C, based on certain features: oocytes in which the MI spindle was rotated by juglone treatment and oocytes in which the spindles were positioned parallel to the cortex were counted as type A; oocytes with weak spindle fibers and unclear spindle poles as type B; and those with abnormal chromosome alignment and distribution as type C. Figure 3D shows the percentage of type A, B, and C oocytes exhibiting irregularities with MI progression and the first polar body extrusion, following juglone treatment. At 5 and 10 µM juglone, 30% and 44% type A, 60% and 39% type B, and 10% and 17% type C oocytes, respectively, were observed. These results establish that Pin1 inhibition affects the spindle structure and intracellular arrangement.

Spindle rotation relies on the actin cytoskeleton and is disrupted by cytochalasin B treatment [23]. The approaching spindles/chromosomes induce the formation of a cortical actin cap and myosin II ring. The cortical actin cap is a structure essential for the extrusion of the polar body [24]. Therefore, to verify the formation of the actin cap, actin filaments (F-actin) were observed using phalloidin staining. Oocytes cultured for 18 h, following treatment with 5 µM juglone, reached MI. Although the actin cap was formed, the spindle was located parallel to the cortex and the oocytes failed to release the first polar body and complete meiosis, unlike control oocytes that released the polar body and matured to MII (Figure 3E). Based on these results, we summarized the causes of hindered polar body extrusion by Pin1 inhibition (Figure 3F). Upon in vitro juglone treatment, the oocytes reached MI, but the spindle rotated while moving to the cell surface and was positioned parallel to the cortex of the oocytes. As a result, although actin caps were formed, polar body extrusion and meiosis could not progress.

### 3.4. Oocytes from Pin1^−/−^ Mice Show Frequent Failure of Extrusion of the First Polar Body

The maturation status of ovulated oocytes was evaluated by assessing α-tubulin localization and the nucleus. Only 23.2% of the ovulated oocytes derived from Pin1^−/−^ mice were morphologically mature (Figure 4). Most oocytes collected from Pin1^−/−^ mice did not exhibit extrusion of the first polar body. On comparing the WT and Pin1^−/−^ mice, the percentages of oocytes in MI were 0% and 50%, in A/TI were 3.8% and 18.6%, in MII were 96.2% and 23.2%, and of those undergoing cell death were 0% and 8.2%, respectively (Figure 4A). In the ovulated oocytes collected from Pin1^−/−^ mice, spindle formation and extrusion of the first polar body were found to be disrupted (Figure 4Ba–i). Typically, we observed incomplete spindle formation (Figure 4Ba), a lack of separation of the cytoplasm and polar body after polar body extrusion (Figure 4Bb,c), and a lack of extrusion of the first polar body outside the cytoplasm (Figure 4Bd–i). Immunoblotting and immunofluorescence staining were used to confirm that these oocytes indeed lacked Pin1 expression (Figure 4C). Taken together, these results show that Pin1 plays an important role in meiosis.

### 3.5. Oocytes from Pin1^−/−^ Mice Do Not Show Specific Localization of Akt and mTOR

We previously reported that Akt and mTOR are involved in spindle formation and polar body extrusion during meiosis in mice [8]. From the immunoblot analysis of MPM2 (Figure 2B), it is expected that Pin1 targets multiple proteins. Because Akt (60 kDa) and mTOR (289 kDa) are candidate proteins based on their molecular weights, the localization of these proteins was analyzed in oocytes derived from Pin1^−/−^ mice. Because Pin1 has been shown to be involved in regulating the phosphorylation of Ser/Thr kinase [12], the localization of total protein as well as of phosphorylated proteins was observed and compared. As Akt and mTOR are localized on the mitotic spindle, their signals were compared by co-staining with α-tubulin. The results showed that localization on the spindle was very weak (Figure 5A–E). Although the signals of both proteins were not completely abolished, the results indicate that Pin1 affected the localization of Akt and mTOR.

### 3.6. Juglone Reduces the Expression of Phospho-Akt and Phospho-mTOR Slightly

Phosphorylation of Akt and mTOR, presumed to be affected by Pin1, was analyzed using immunoblotting to determine the levels of these proteins. To examine the inhibitory effect of Pin1 in oocytes, the cells were cultured for 18 h in medium supplemented with juglone and subjected to immunoblot analysis. The levels of phosphorylated Akt and phosphorylated mTOR tended to be reduced by juglone (Figure 6). 

## 4. Discussion

Pin1 is widely expressed in somatic cells and plays a central role in coordinating and controlling the cell cycle, which is critical to the survival of most tissues [12,13]. Disruption in the balance of Pin1 activation can lead to diseases, including cancer and Alzheimer’s disease [25,26]. In mouse ovaries, *Pin1* mRNA is expressed via a gonadotropin-dependent mechanism [27]. Furthermore, follicle-stimulating hormone has been reported to increase the expression of *Pin1* mRNA in granulosa cells in cows [28]. However, there are no reports on the expression and function of Pin1 in oocytes. Based on its function in somatic cells and its expression in the ovaries, Pin1 is believed to be involved in meiotic maturation. MPM2 is used as a marker of the mitotic index and the phosphorylated proteins that are detected by this antibody could also be potential targets of Pin1 [29,30]. Accordingly, in this study, we examined the role of Pin1 in mouse oocytes by evaluating the expression and localization of Pin1 and MPM2 during oocyte maturation.

We show that Pin1 is continuously expressed in the cytoplasm during meiotic maturation, and its localization is altered with meiotic progression. The localization of Pin1 was observed to be similar to that of phospho-mitotic proteins detected using the anti-MPM2 antibody, suggesting that Pin1 functions in oocytes. In the GV stage, Pin1 was localized both in the cytoplasm and at the nuclear membrane in a punctate manner, as were the proteins detected using the anti-MPM2 antibody. These results suggest that the localization of these two proteins is associated with mTOR and pericentrin, an integral component of the centrosome fibrous matrix that forms the spindle poles [8]. 

The nuclear membrane is composed of lamins and membrane proteins, and nuclear lamins are phosphorylated by CDC2 kinase, thereby, inducing GVBD [31,32]. A study using HCMV-infected fibroblasts showed that Pin1 binds to pSer22-Pro of lamin A and promotes the disintegration of the nuclear membrane by modulating the conformation of lamin [33]. Furthermore, it has been reported that Pin1 is essential for the maintenance of the structure of nuclear B-type lamin and the anchoring function of heterochromatin protein 1α, playing a role in preventing alterations in the nuclear membrane and relaxation of the heterochromatin [34]. Based on these reports, we speculated that Pin1 binds to phosphorylated lamins and induces and promotes the disruption of the nuclear membrane in mouse oocytes. However, in this study, juglone treatment in vitro did not block oocytes at GV. This suggests that different pathways excluding Pin1 strongly support the collapse of the nuclear envelope membrane or that the effects of juglone are not sufficient.

After GVBD, the spindle is formed, and the polar body is extruded for meiosis to progress. Formation of the meiotic spindle and movement of the spindle toward the cell surface for asymmetric division, are strictly controlled by the microfilament and actin network [35]. We found that Pin1 was expressed in the cytoplasm of oocytes after GVBD, whereas the expression of the proteins detected using the anti-MPM2 antibody increased rapidly after GVBD and was localized at the cell surface where actin filaments converge. This suggests that Pin1 affects actin and its related factors. Moreover, Pin1 was localized on the meiotic spindle in the MI and MII stages and on the contractile ring in A/TI. This localization strongly indicates that Pin1 plays an important role in spindle formation and the extrusion of the first polar body. Interestingly, localization of Pin1 on the meiotic spindle was observed only with fresh MII oocytes. This localization disappeared in oocytes (24 h after hCG injection) at certain time points after the extrusion of the polar body and was instead observed at the periphery of the aligned metaphase chromosome. At this time, because anti-MPM2-reactive proteins were localized on the spindle, it became clear that Pin1 disappeared from the spindle prior to oocyte aging. This supports previous findings suggesting that Pin1 may be involved in the progression of the cell cycle network as well as act as an indicator of fertilization as a molecular timer in oocytes [12]. 

MPM2 specifically recognizes phosphorylated proteins during mitosis. In mouse oocytes, we confirmed the presence of a 100 kDa or heavier protein at 8 h after IVM using immunoblot analysis. During this time, more than 70% of oocytes exhibited GVBD or were in MI, suggesting that multiple phosphorylated proteins are involved in meiosis as targets of Pin1. There were no significant differences in the expression patterns of Pin1 and anti-MPM2-reactive phospho-proteins in IVM-matured oocytes and ovulated (in vivo-matured) oocytes; however, the levels of these proteins were higher in in vivo-matured oocytes.

To examine the role of Pin1 in oocyte maturation, IVM was performed using the Pin1 inhibitor juglone for 18 h. We found that the addition of juglone suppressed the extrusion of the first polar body, and the number of oocytes reaching the MII phase decreased in a dose-dependent manner. In oocytes not showing extrusion of the polar body, incomplete MI spindle formation and chromosome separation within the cells were observed. Formin-2 is involved in spindle formation and asymmetric division in oocytes [36,37] and is localized on the meiotic spindle. A deficiency of Formin-2 suppresses the extrusion of the polar body. Juglone treatment induces a nuclear phase similar to that observed in Formin-2-deficient oocytes, suggesting that Formin-2 may be a target of Pin1 in meiotic maturation [36]. In mammals, the migration of the meiotic I spindle to a subcortical location is a crucial step in ensuring asymmetric cell division during the extrusion of the first polar body. MI spindle migration is an actin cytoskeleton-dependent process and is blocked by the inhibition of actin polymerization [38]. Spindle migration also relies on the activity of myosin, an actin-associated motor protein. Myosin is activated by myosin light chain kinase, the inhibition of which disrupts spindle migration [39]. When juglone treatment was performed in vitro, the oocytes reached MI, but the spindle rotated while moving to the cell surface and was positioned parallel to the cortex of the oocytes. As a result, actin caps were formed but extrusion of the polar body and meiosis could not progress. RhoA, a small GTPase, regulates spindle rotation through the organization of F-actin; however, the exact molecular mechanism underlying this regulation remains unknown [19]. Although mechanisms regulating spindle rotation are unclear, our results suggest that a Pin1-mediated signaling pathway is involved in spindle rotation. 

Pin1 was found to be localized on the spindle; however, its localization was reduced over time. We found that Akt and mTOR were involved in spindle formation and polar body extrusion in mouse oocytes, and the changes in localization were consistent with these findings [6,8,17]. It has been reported that, in rat adipocytes, the phosphorylation site of mTOR matches the Ser/Thr-Pro motif and that Pro is a determinant of mTOR specificity. These reports suggest that even during meiosis, mTOR is a target of Pin1 [40]. In mouse oocytes, Pin1 inhibition reduced the localization of Akt and mTOR in the cytoplasm, particularly on the spindle, whereas slight expression was observed in knockout mouse-derived oocytes. In humans, phosphorylation at Thr92-Pro and Thr450-Pro motifs is important for maintaining the stability and activation of Akt, and Pin1 binds to the Ser/Thr-Pro motifs of Akt. Because the Thr92-Pro and Thr450-Pro motifs are highly conserved in various Akt isoforms across species, in mouse oocytes, Pin1 binds to the Ser/Thr-Pro motifs of Akt, and it is speculated that it is involved in maintenance of Akt stability [41]. 

In Pin1^−/−^ mice, more than 78% of the ovulated oocytes did not exhibit polar body extrusion, were arrested at MI or A/TI, and typically presented with incomplete spindle formation. Pfender et al. reported that spire1 and spire2 cooperate with Formin-2 to form nuclei in the actin filaments of mouse oocytes and are essential for organization of actin into contracted rings during polar body extrusion [42]. To accurately separate the number of chromosomes, the cytoplasm and polar body are separated by the contractile rings after extrusion of the polar body; however, the function of the contractile ring was disrupted in Pin1^−/−^ oocytes. Moreover, the lack of polar body extrusion after chromosome separation may be related to the function of myosin II, because the failure of Pin1^−/−^ oocytes in releasing the polar body is similar to that seen in the case of myosin II inhibition with ML-7 and blebbistatin [43].

Multiple mRNAs and proteins are involved in oocyte maturation, fertilization, and early embryonic development. These mRNAs and proteins are biosynthesized and stored during development after the oocyte arrest at the diplotene stage of prophase of meiosis. During this time, oocyte transcriptional activity is silenced, but the maturation process is driven by post-transcriptional regulation of mRNAs that are conserved within the oocyte (maternal mRNA).

The expression of several maternal mRNAs is regulated by cytoplasmic polyadenylation. Cytoplasmic polyadenylation is a widespread phenomenon that controls the translation and stability of mRNAs. In *Xenopus* oocytes, polyadenylation is controlled by CPEB, a sequence-specific RNA-binding protein [44]. During this meiotic transition, CPEB is subjected to phosphorylation-dependent ubiquitination and partial destruction, which is necessary for successive waves of polyadenylation of distinct mRNAs [45]. Pin1 was identified as the factor that catalyzes this CPEB disruption and has been reported to be a key factor regulating CPEB degradation during the maturation of *Xenopus* oocytes [45]. In mice, the transition to MI and MII requires M phase-promoting factor (MPF: cdc2 and cyclin B) activity. This increase in activity is mainly dependent on the cytoplasmic polyadenylation of cyclin B1 mRNA [46]. Most of the mRNAs bound to translational ribosomes in mouse oocytes contain motifs for the RNA-binding proteins, CPEB1 and DAZL (deleted in azoospermia-like) [47]. CPEB1 activates cytoplasmic polyadenylation of Dazl mRNA and promotes translation during the transition from the GV stage to the MI stage before the onset of oocyte maturation. Thereafter, CPEB1 is degraded from the MI stage to the MII stage but the translation of target mRNAs, including that of Dazl mRNA, by DAZL is promoted, and oocyte maturation proceeds normally [47]. Because of its broad range of target proteins, Pin1 functions in diverse cellular processes, such as cell cycle control, transcription, splicing regulation, DNA replication, and DNA damage response [48]. Pin1 may also be involved in regulating CPEB in oocyte maturation in mice.

Although meiosis is regulated by a complex mechanism mediated by many kinase proteins, Pin1 may function to balance the mechanism. Overall, these findings suggest that Pin1 is involved in modulating cytoskeleton structure and function during oocyte maturation. However, further investigations are required to determine the other target proteins of Pin1.

## 5. Conclusions

We show that Pin1 is continuously expressed in mouse oocytes and facilitates meiotic maturation by regulating multiple target proteins. To the best of our knowledge, this study is the first to describe the role of Pin1 in the meiotic maturation of oocytes. Our study provides valuable insights into the functions of this protein as a critical factor in controlling intracellular events. We postulate that the functional regulation of Pin1 is indispensable in oocyte production and should be considered for the development of oocyte culture technologies in reproductive medicine and animal breeding.

## Figures and Tables

**Figure 1 cells-11-03772-f001:**
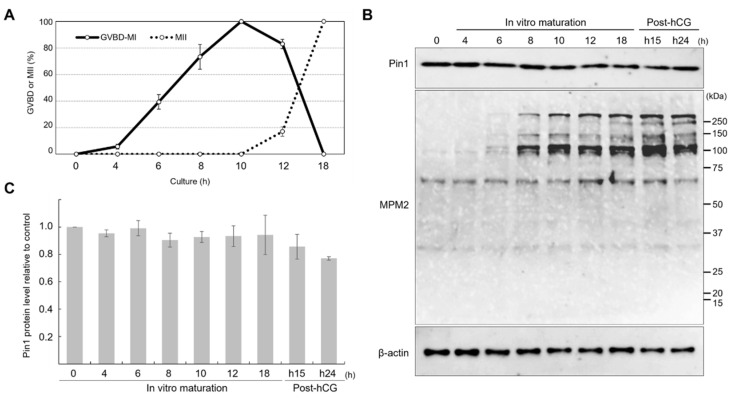
In vitro maturation of oocytes and expression of Pin1 and phosphorylated mitotic proteins detected using anti-MPM2 antibody at different stages of maturation in vitro and in vivo. (**A**) Maturation rates at different times during in vitro maturation of oocytes. Solid and dashed lines indicate the ratios of oocytes in germinal vesicle breakdown-metaphase I (GVBD-MI) and metaphase II (MII), respectively. Data are expressed as means ± SEM for five independent experiments. (**B**) Immunoblot analysis using anti-Pin1 and anti-MPM2 antibodies and extracts of oocytes collected at 0, 4, 6, 8, 10, 12, and 18 h after the start of in vitro culture and in vivo-matured oocytes collected from the oviducts of mice at 15 (h15) and 24 h (h24) after human chorionic gonadotropin injection. Extracts of 50 oocytes were applied to each lane. β-actin was used as the loading control. Immunoblot analysis was performed using five sets to confirm reproducibility. (**C**) Protein levels of Pin1. β-actin was used as a reference for quantification. The values shown are means ± SEM of three biological replicates. The expression level at 0 h is 1; changes in the expression level of Pin1 over time are shown.

**Figure 2 cells-11-03772-f002:**
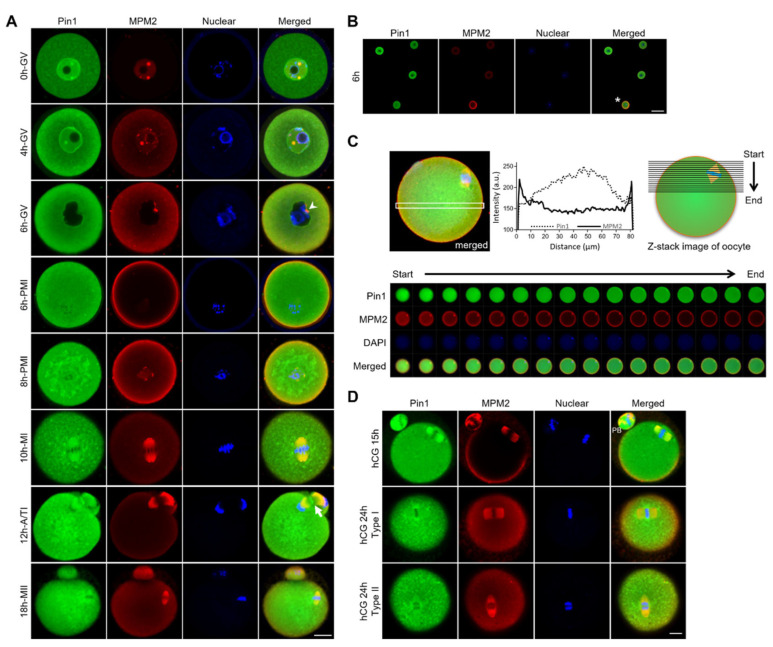
Pin1 and anti-MPM2-reactive phospho-mitotic proteins are dynamically localized during meiotic maturation. (**A**) Localization of Pin1 and phospho-proteins detected using anti-MPM2 antibody at different stages of oocytes collected at 0, 4, 6, 8, 10, 12, and 18 h after the start of in vitro maturation. The panels show typical localization patterns at germinal vesicle (GV), pro-metaphase I (PMI), metaphase I (MI), anaphase/telophase I (A/TI), and metaphase (MII) stages. Arrowheads indicate distortion of the nuclear membrane (6 h GV), and arrows indicate localization of Pin1 in the midbody at A/TI (12 h A/TI). Scale bar, 20 µm. (**B**) Comparison of the fluorescence intensities of Pin1 and anti-MPM2-reactive phospho-mitotic proteins in the cytoplasm between GV and immediately after germinal vesicle breakdown (GVBD). The panels show oocytes at 6 h after the start of culture; four oocytes are shown at low magnification. The upper three oocytes are in the GV stage, and one oocyte (indicated with an asterisk) has just after GVBD. Scale bar, 100 µm. (**C**) Comparison of the localization of anti-MPM2-reactive phospho-mitotic proteins and Pin1 in the M phase. The graph shows the fluorescence intensity of Pin1 and MPM2 in the white-framed area of the merged image. The panels show continuous photographs obtained using the Z-stack analysis from the cell surface layer toward the cell center in MI oocytes. (**D**) Changes in the localization of Pin1 and MPM2 antigens during the aging of oocytes. The panels show typical localization patterns in mature oocytes collected from oviducts at 15 and 24 h after hCG injection. Green, red, and blue show Pin1, anti-MPM2-reactive phospho-mitotic proteins, and nuclei, respectively. Scale bar, 20 µm. PB, polar body.

**Figure 3 cells-11-03772-f003:**
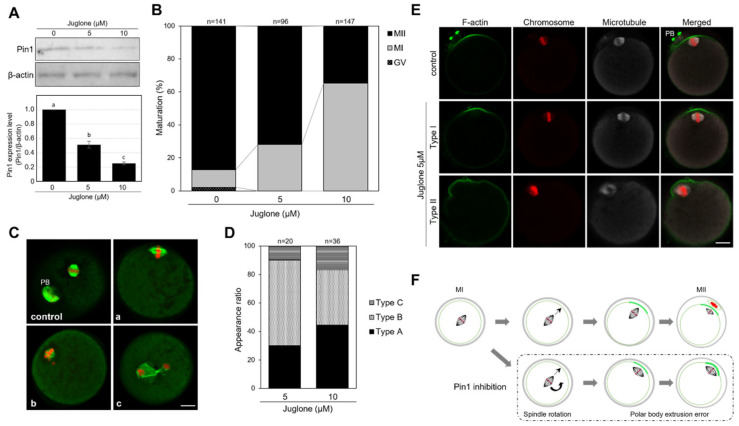
Juglone inhibits meiotic maturation in vitro. (**A**) Expression level of Pin1 after 18 h of juglone treatment. (**B**) Effects of Pin1 inhibition with juglone on meiotic maturation as determined using immunostaining with anti-α-tubulin antibodies and nuclear staining. Graphs showing the percentage of oocytes, with or without juglone treatment, at the germinal vesicle (GV), metaphase I (MI), and metaphase II (MII) stages at 18 h after in vitro maturation (IVM). The inhibition experiment was repeated five times. Graphs show the ratios of the total number. (**C**) Localization of α-tubulin and chromosomes after 18 h of juglone treatment. A photograph of oocytes cultured without inhibitor for 18 h is shown as a control. Most oocytes, after juglone treatment, appeared to be arrested at MI at 18 h after the start of culture with MI spindles parallel to the cortex, weak spindle fibers, and unclear spindle poles (a: type A). Spindles are small, and chromosomes are scattered away from the metaphase plate (b: type B). The polar body is not extruded, and chromosomes are separated in the cytoplasm (c: type C). Green and red indicate α-tubulin and chromosomes, respectively. Scale bar, 20 µm. PB, polar body. (**D**) Graphs showing the ratios of Type A, B, and C oocytes that stopped at the MI phase and could not extrude the first polar body following juglone treatment at 18 h after IVM. (**E**) Localization of actin cap using F-actin staining, 18 h after IVM. The control oocytes show polar body extrusion and strong localization of actin near the MII spindle. After treatment with 5 µM juglone, the spindle is localized to the nearest site on the cortex, but the first polar body is not extruded even with the formation of the actin cap. In type I oocytes, the MI spindle appeared parallel to the cortex. In type II oocytes, the rotated spindle did not release the polar body even after a few hours. Green, red, and gray indicate F-actin, chromosomes, and microtubules, respectively. (**F**) Schematic diagram of the effect of juglone treatment on meiosis. The top panel in the figure shows meiosis from normal MI to MII. The lower panel shows the effect of Pin1 inhibition. When Pin1 is inhibited by juglone, the spindle in the MI moves to the cortex but the spindle rotates and is placed parallel to the cortex, such that the release of the polar body is not possible even when an actin cap is formed. Green and red indicate F-actin and chromosome, respectively. Scale bar, 20 µm. Data represent the mean ± SEM of three replicates repeated in different experiments. Different letters represent significant difference for *p* < 0.05 between treatments.

**Figure 4 cells-11-03772-f004:**
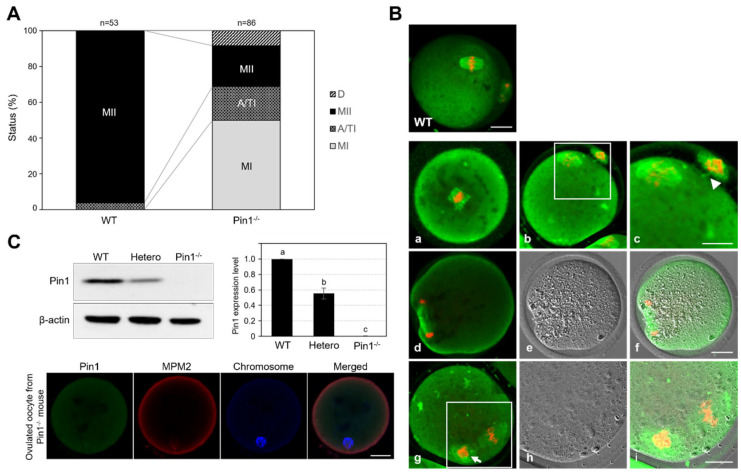
Most oocytes from Pin1^−/−^ mice did not show meiosis progression. (**A**) Graph showing the stages of ovulated oocytes determined by the localization of α-tubulin and nuclear status. MI, metaphase I; A/TI, anaphase/telophase I; MII, metaphase II; D, cell death. (**B**) Typical localization patterns of α-tubulin and nuclei in Pin1^−/−^ oocytes. An image of oocytes collected from wild-type (WT) mice is shown as a control. The oocytes from Pin1^−/−^ mice shown in panels (**a**), (**d**), and (**g**) are arrested at MI or do not show proper extrusion of the first polar body. The oocytes shown in panel (**b**) show proper extrusion of the first polar body, but the polar body and the cytoplasm are not separated, and the morphology of the spindles and chromosomes in the oocytes is not clear. Arrowheads indicate the cytoplasm. The inset in panel (**c**) is indicated by a white box in panel (**b**). Panel (**e**) shows a bright-field image and (**f**) shows the bright-field image merged with the fluorescence image from panel (**d**). Panels (**h**) and (**i**) are enlarged portions from panel (**g**), showing a bright-field image and the bright-field image merged with a fluorescence image, respectively. The inset is indicated by a white box in panel (**g**). Arrows indicate segregated chromosomes in the cytoplasm. Scale bar, 20 µm. Green and red indicate α-tubulin and chromosomes, respectively. (**C**) Immunoblotting analysis of the expression of Pin1 in oocytes from WT, heterozygous, and Pin1^−/−^ mice; no expression of Pin1 is seen in oocytes from Pin1^−/−^ mice. The graph shows the levels of Pin1 normalized to that of β-actin. Immunofluorescence staining shows significantly reduced expression of Pin1 and phospho-mitotic proteins detected using the anti-MPM2 antibody in oocytes from Pin1^−/−^ mice. Green, red, and blue indicate Pin1, MPM2, and nuclei, respectively. Scale bar, 20 µm. Data represent the mean ± SEM of three replicates repeated in different experiments. Different letters represent significant difference for *p* < 0.05.

**Figure 5 cells-11-03772-f005:**
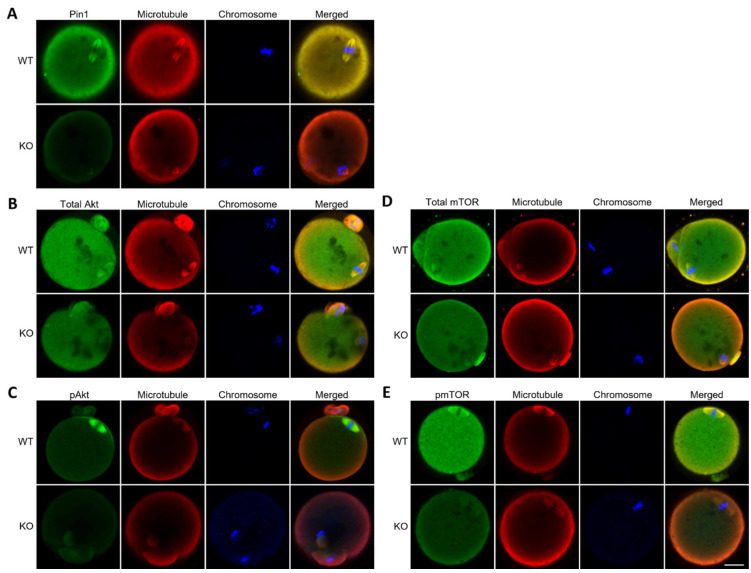
Pin1 deficiency inhibits the localization of Akt and mTOR in the oocytes. Akt and mTOR are serine/threonine kinases and are candidate targets of Pin1. The localization of total Akt, phospho-Akt (pAkt, Ser473), total mTOR, and phospho-mTOR (pmTOR, Ser2448) was analyzed in the ovulated oocytes from wild-type (WT) and Pin1 knockout (KO) mice. Green indicates (**A**) Pin1, (**B**) total Akt, (**C**) pAkt, (**D**) total mTOR, or (**E**) pmTOR. Red and blue indicate microtubules and chromosomes, respectively. Scale bar, 20 µm.

**Figure 6 cells-11-03772-f006:**
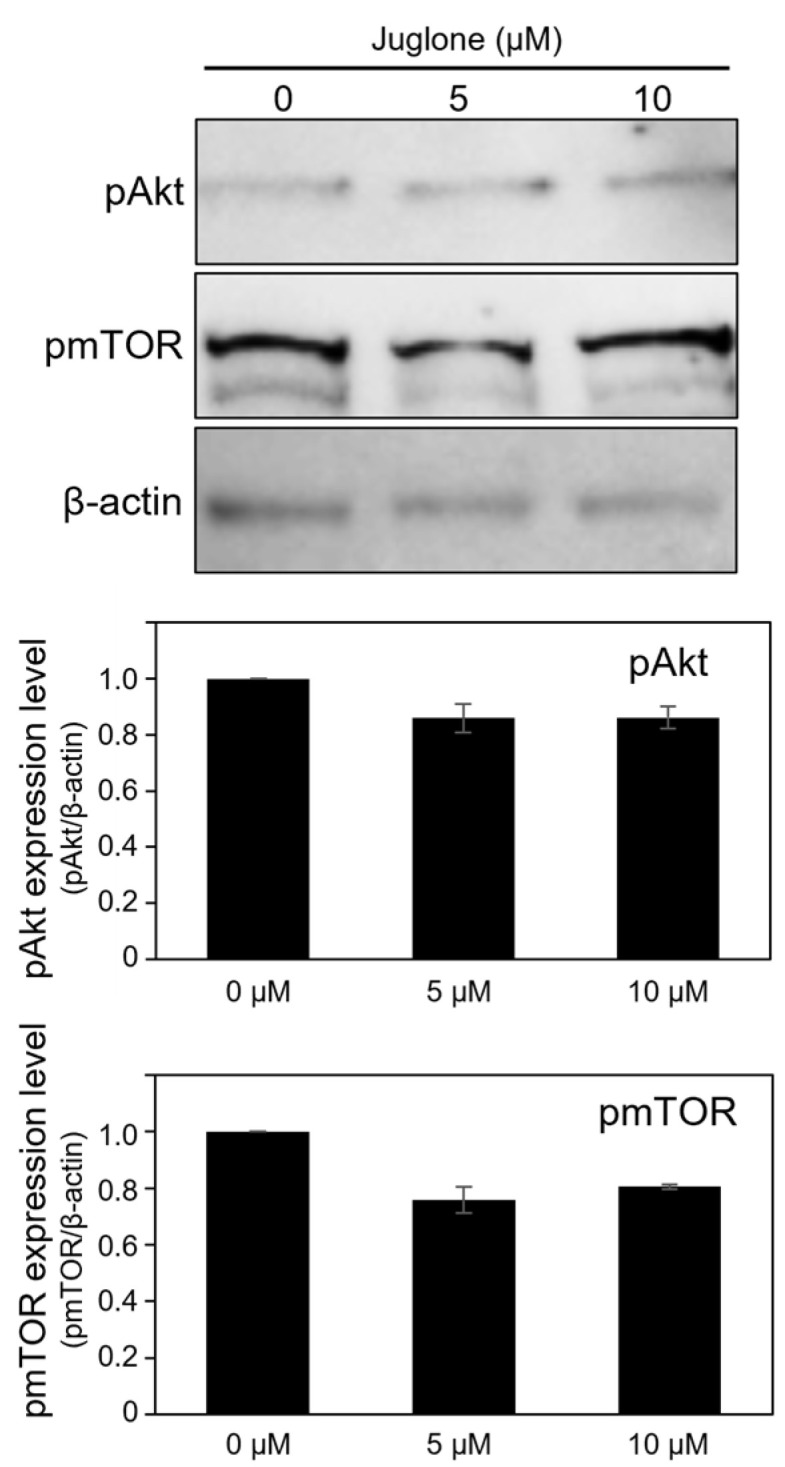
Pin1 inhibition in in vitro maturation (IVM) slightly decreased the expression of pAkt and pmTOR in the oocytes. Fifty oocytes that underwent 18 h IVM in medium supplemented with 0, 5, or 10 µM juglone were subjected to immunoblot analysis. The expression level was normalized to β-actin and quantified. Data represent the mean ± SEM of three replicates repeated in different experiments.

## Data Availability

Not applicable.

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
