# Peer review of "Prolyl Isomerase, Pin1, Controls Meiotic Progression in Mouse Oocytes"

_cells, 2022, doi:10.3390/cells11233772_

Round 1

Reviewer 1 Report (Previous Reviewer 1)

In this version of the manuscript, the authors included additional experiments, focused on changes of the localization of Akt and mTOR in ovulated MII Pin1 +/+ and -/- oocytes.

My questions are following:

Are the authors suggesting that the changes in the localization of these two proteins are responsible for the lower PBE rate in Pin1 deficient oocytes or oocytes treated with the inhibitor? Can this phenotype (localization and perhaps also PBE) be rescued by overexpression of Pin1?

Would it be possible to quantify the changes in the localization of both proteins, together with changes in the level of the total protein in Pin1 deficient oocytes?

Author Response

Reviewer 2 Report (Previous Reviewer 2)

The key to the formation of fertilized eggs lies in the combination of mature oocytes and sperm cells. Then oocyte maturation is crucial. This study is novel in the necessity of Pin1 for oocyte meiotic maturation. The authors explored the expression and localization of Pin in different stages of oocyte maturation, applied the inhibitor juglone to affect oocyte polar body discharge, and found that after Pin knockout, oocyte polar body discharge was affected. In short, the article has some points that deserve further improvement. 

Was the fluorescence experiment repeated three times? How many cells were used in each fluorescence experiment? Adding fluorescence intensity data to the histogram is more intuitive.

How many mice were used should be mentioned in the mice experiments.

The data statistical analysis method used in the experiment should be mentioned in Materials and Methods section.

Figure 6 Protein experiments, should be repeated more than three times, if repeated, it would be better to add the average to the bar graph.

There are numerous issues throughout the manuscript regarding grammar, word choice, expression choice, scientific tone, and overall flow of the text should be carefully check.

Round 2

Reviewer 1 Report (Previous Reviewer 1)

In their response, authors did not answer my question related to the role of Akt and mTOR in PBE in oocytes. They obviously do not plan to carry out rescue experiment (microinjection of Pin1 into GV oocytes for example) and to quantify potential changes in the expression and localization of Akt and mTOR in Pin1 deficient oocytes. It is therefore unclear to me, why they have included Akt and mTOR results in the manuscript.  

My concern is that without better understanding of the function of Pin1 at the molecular level, the results in this manuscript are mostly observations. In my opinion the authors missed opportunity to obtain knowledge about molecular mechanisms regulating meiotic division in mammals. I do not recommend the manuscript for the publication in Cells journal.

This manuscript is a resubmission of an earlier submission. The following is a list of the peer review reports and author responses from that submission.

Round 1

Reviewer 1 Report

Manuscript “Prolyl isomerase, Pin1, controls meiotic progression in mouse 2

Oocytes” by Hoshino and Uchida focuses on the role of Pin1 in mouse oocytes during meiotic maturation. Authors studied the protein levels and localization pattern of Pin1, and also used specific inhibitor and transgenic technology to block Pin1 function.

Although it seems clear that Pin1 is required for mouse female meiotic maturation, as both Pin1 inhibition and knockout strongly suggest that, authors failed to provide any data about the actual function of this molecule in mouse meiosis I. I believe that more experimental work is necessary to document Pin1 potential role in GVBD, spindle positioning and PBE.  Authors should pay attention to previously published report from Xenopus (Nechama et al., 2013), in which Pin1 seems to have a role in control of CPEB levels, and therefore translation of important molecules, including cyclin B, during meiosis. Such function would be in accordance to the phenotype presented here.

Overall, the study is very descriptive and lacks molecular insight into Pin1 role in mouse meiosis. Therefore, the I do not recommend this study for publication in Cells journal. 

Reviewer 2 Report

This study is novel in the necessity of Pin1 for oocyte meiotic maturation. The author explored the expression and localization of Pin in different stages of oocyte maturation, applied the inhibitor juglone to affect oocyte polar body discharge, and found that after Pin knockout, oocyte polar body discharge was affected. In short, the article has some points that deserve further improvement.

1. Figure 1 B lacks a graph of protein levels.

2. Figure 2 is missing BCD.

3. The order of pictures in Figure 3 needs to be adjusted.

4. The inhibitory effect of Pin1 inhibitor juglone (5-hydroxy-1,4-naphthoquinone) lacks validation experiments, such as protein level or RNA level.

5. What specific pathway and mechanism does Pin1 affect meiotic maturation? There is no data to support it in the article. More experiments should be carried out to illustrate.